# Control of Healthcare-Associated Carbapenem-Resistant *Acinetobacter baumannii* by Enhancement of Infection Control Measures

**DOI:** 10.3390/antibiotics11081076

**Published:** 2022-08-08

**Authors:** Shuk-Ching Wong, Pui-Hing Chau, Simon Yung-Chun So, Germaine Kit-Ming Lam, Veronica Wing-Man Chan, Lithia Lai-Ha Yuen, Christine Ho-Yan Au Yeung, Jonathan Hon-Kwan Chen, Pak-Leung Ho, Kwok-Yung Yuen, Vincent Chi-Chung Cheng

**Affiliations:** 1Infection Control Team, Queen Mary Hospital, Hong Kong West Cluster, Hong Kong SAR, China; 2School of Nursing, Li Ka Shing Faculty of Medicine, The University of Hong Kong, Hong Kong SAR, China; 3Department of Microbiology, Queen Mary Hospital, Hong Kong SAR, China; 4Department of Microbiology, Li Ka Shing Faculty of Medicine, The University of Hong Kong, Hong Kong SAR, China

**Keywords:** carbapenem-resistant *Acinetobacter baumannii*, multidrug-resistant *Acinetobacter baumannii*, carbapenem-resistant Enterobacterales, healthcare-associated infection, antimicrobial consumption, infection control, hand hygiene, directly observed hand hygiene

## Abstract

Antimicrobial stewardship and infection control measures are equally important in the control of antimicrobial-resistant organisms. We conducted a retrospective analysis of the incidence rate of hospital-onset carbapenem-resistant *Acinetobacter baumannii* (CRAB) infection (per 1000 patient days) in the Queen Mary Hospital, a 1700-bed, university-affiliated teaching hospital, from period 1 (1 January 2007 to 31 December 2013) to period 2 (1 January 2014 to 31 December 2019), where enhanced infection control measures, including directly observed hand hygiene before meal and medication rounds to conscious patients, and the priority use of single room isolation, were implemented during period 2. This study aimed to investigate the association between enhanced infection control measures and changes in the trend in the incidence rate of hospital-onset CRAB infection. Antimicrobial consumption (defined daily dose per 1000 patient days) was monitored. Interrupted time series, in particular segmented Poisson regression, was used. The hospital-onset CRAB infection increased by 21.3% per year [relative risk (RR): 1.213, 95% confidence interval (CI): 1.162–1.266, *p* < 0.001], whereas the consumption of the extended spectrum betalactam-betalactamase inhibitor (BLBI) combination and cephalosporins increased by 11.2% per year (RR: 1.112, 95% CI: 1.102–1.122, *p* < 0.001) and 4.2% per year (RR: 1.042, 95% CI: 1.028–1.056, *p* < 0.001), respectively, in period 1. With enhanced infection control measures, the hospital-onset CRAB infection decreased by 9.8% per year (RR: 0.902, 95% CI: 0.854–0.953, *p* < 0.001), whereas the consumption of the extended spectrum BLBI combination and cephalosporins increased by 3.8% per year (RR: 1.038, 95% CI: 1.033–1.044, *p* < 0.001) and 7.6% per year (RR: 1.076, 95% CI: 1.056–1.097, *p* < 0.001), respectively, in period 2. The consumption of carbapenems increased by 8.4% per year (RR: 1.84, 95% CI: 1.073–1.094, *p* < 0.001) in both period 1 and period 2. The control of healthcare-associated CRAB could be achieved by infection control measures with an emphasis on directly observed hand hygiene, despite an increasing trend of antimicrobial consumption.

## 1. Introduction

Carbapenem-resistant *Acinetobacter baumannii* (CRAB) is an emerging healthcare-associated pathogen [1,2]. Nosocomial transmission and hospital outbreaks, especially in the neonatal and adult intensive care units (ICUs), have been reported in the United States, Greece, Vietnam, and Singapore [3,4,5,6]. A recent systematic review and meta-analysis also showed that CRAB accounted for 13.6% of all hospital-acquired infections in ICUs across Europe, the Eastern Mediterranean, and Africa [7]. The high prevalence of CRAB in ICUs could be attributed to the use of broad-spectrum antimicrobial agents in critically ill patients [7]. Given the heterogeneity of antimicrobial consumption in acute care hospitals [8], the correlation between antimicrobial consumption and the prevalence of CRAB has been observed [9,10,11]. Mathematical models also suggest the existence of thresholds of antimicrobial consumption beyond which resistance would be triggered [12]. The antimicrobial stewardship program was associated with a significant impact in reducing the incidence of CRAB in the hospital [13] and contributed to the control of outbreak [14].

Infection control measures other than the antimicrobial stewardship program have been demonstrated to control the emerging CRAB in hospitals during outbreak [3,4,15,16,17,18] and non-outbreak periods [19,20,21,22,23,24,25,26,27,28]. These measures include enforcing hand hygiene practice, the education of hospital staff, single-room isolation or the cohorting of infected patients, environmental cleaning and disinfection, active surveillance culture upon admission and during hospitalization, daily chlorhexidine bathing, and the renovation of a multi-bed bay room into single rooms [3,4,15,16,17,18,19,20,21,22,23,24,25,26,27,28]. 

In our institution, an innovative measure of directly observed hand hygiene-based infection control has been promoted since 2014 [29]. Directly observed hand hygiene was carried out by a healthcare assistant, who delivered alcohol-based hand rub to the conscious hospitalized patients before meal and medication rounds to reduce the risk of oral acquisition of antimicrobial-resistant organisms [30]. Directly observed hand hygiene-based infection control measures successfully prevented or controlled the transmission of epidemiologically important viruses [31,32,33] and antimicrobial-resistant organisms in the hospitals [34,35,36]. With the implementation of directly observed hand hygiene-based infection control measures over 5 years, we seek to demonstrate the effect of this measure on the incidence rate of CRAB, together with the other antimicrobial-resistant gram-negative organisms, including multidrug-resistant *Acinetobacter baumannii* (MRAB), carbapenem-resistant Enterobacterales (CRE), and cephalosporin-resistant Enterobacterales (CephRE). 

## 2. Materials and Methods 

### 2.1. Setting

This is a retrospective study on the epidemiology of healthcare-associated CRAB in the Queen Mary Hospital, a 1700-bed, university-affiliated teaching hospital providing a tertiary referral service for medical and surgical procedures, orthopedics, pediatrics, obstetrics and gynecology, oncology, and solid organ and bone marrow transplantation in Hong Kong. The study period is divided into two parts, where period 1 (1 January 2007 to 31 December 2013) is the baseline and period 2 (1 January 2014 to 31 December 2019) is the intervention, with the enhancement in the infection control measures. 

### 2.2. Promotion of Hand Hygiene and Directly Observed Hand hygiene 

Hand hygiene using alcohol-based hand rub has been implemented since 2007, and the Queen Mary Hospital was the pilot center of the hand hygiene campaign by the World Health Organization (WHO). WHO formulation I (ethanol 80% v/v, glycerol 1.45% v/v, hydrogen peroxide 0.125% v/v), and formulation II (isopropyl alcohol 75% v/v, glycerol 1.45% v/v, hydrogen peroxide 0.125% v/v) alcohol-based hand rubs were promoted to our healthcare workers, according to “My Five Moments for Hand Hygiene.” Hand hygiene has been further promoted by healthcare workers to the hospitalized patients since 2013. In addition to the promotion of patient empowerment in hand hygiene [37], directly observed hand hygiene before meal and medication rounds, as a part of an enhancement in infection control measures for conscious hospitalized patients to reduce the risk of the oral acquisition of antimicrobial-resistant organisms, was fully implemented in 2014 [29,30]. Initially, ward nurses were responsible for the implementation of directly observed hand hygiene by delivering 3 mL of alcohol-based hand rub to the conscious patients before meal and medication rounds. In 2015, the healthcare assistants in each ward were appointed as the hand hygiene ambassadors to carry out this designated task of directly observed hand hygiene to patients [30].

Unobtrusive hand hygiene audits to monitor the hand hygiene compliance among healthcare workers were performed by infection control nurses, according to recommendations of the WHO [38]. The practice of directly observed hand hygiene of hospitalized patients was also monitored by infection control nurses. Three conscious and communicable hospitalized patients were randomly selected in each ward at least once weekly to enquire if the healthcare workers had delivered alcohol-based hand rub to them before meal and medication rounds. Compliance with the directly observed hand hygiene in the ward was defined if all three or two out of three patients confirmed that alcohol-based hand rub had been delivered to them before meal and medication rounds.

### 2.3. Infection Control Measures for Antimicrobial-Resistant Organisms

In general, standard infection control measures, including hand hygiene, cohort nursing, and environmental disinfection, were performed in both period 1 and period 2. Directly observed hand hygiene was implemented in period 2. For the control of various antimicrobial-resistant gram-negative organisms, infection control nurses reviewed the microbiology laboratory database for patients with cultures positive for CRAB, MRAB, CRE, and CephRE, and recommended appropriate infection control measures. In addition to the directly observed hand hygiene measures, the priority use of single-room isolation was applied to patients cultured positive for CRAB and MRAB in period 2. Due to the limited number of isolation rooms, contact precautions with respect to cohort nursing were implemented for patients cultured positive for CRE. In view of the high prevalence of CephRE, standard precautions were used for patients cultured positive for CephRE. 

### 2.4. Data Source

The episode-based records of all inpatients were retrieved from the Clinical Data Analysis and Reporting System (CDARS), an electronic database of health records under the governance of the Hospital Authority, as previously described [39]. Briefly, a unique hospital number was assigned to each hospital admission. The duration of hospitalization for each admission was recorded. Episodes of each hospital admission involving a positive culture for the selected microorganisms from the clinical specimens over 13 years (from 1 January 2007 to 31 December 2019) were retrieved. The selected microorganisms included *Acinetobacter baumannii, Escherichia coli, Klebsiella species*, and *Enterobacter species*, together with their antimicrobial susceptibility pattern. Antimicrobial susceptibility was defined according to the recommendations of the Clinical and Laboratory Standards Institute [40]. 

Antimicrobial consumption data during the study period were also retrieved from the CDARS. The antimicrobial consumption data of all selected classes of antimicrobial agents were analyzed by year and expressed as the defined daily dose (DDD) per 1000 patient days.

### 2.5. Trends of Antimicrobial-Resistant Organisms

The incidence rates of the selected antimicrobial-resistant organisms, including CRAB, MRAB, CRE, and CephRE, expressed as 1000 patient days from 2007 to 2019, were analyzed. The incidence rates of CRE and CephRE were analyzed as the internal controls, because the priority use of single-room isolation was not applied to patients infected with CRE and CephRE. The laboratory specimens collected for clinical purposes were counted, while only the specimens collected for active surveillance testing purposes were excluded, according to the laboratory-identified event reporting measures defined by the National Healthcare Safety Network (NHSN) of Centers for Disease Control and Prevention (CDC) of the United States [41].

### 2.6. Hospital-onset Events before and after the Enhancement of Infection Control Measures 

Among our hospitalized patients, an episode of a hospital-onset event of antimicrobial-resistant organism infection was defined as the clinical specimen being cultured positive for the antimicrobial-resistant organism, collected on or after hospital day 4 [41]. If a patient had more than one episode of a hospital-onset event due to the same pathogen with the same hospital number, only the first episode was recorded, regardless of the specimen source. The incidence rates of these events were expressed as the number of episodes per 1000 patient days.

CRAB is usually multidrug-resistant [42]. The incidence rate of hospital-onset CARB was used as a proxy to evaluate the outcome of the enhancement of infection control measures, with directly observed hand hygiene before meal and medication rounds together with the priority use of single-room isolation, which was implemented during period 2. The incidence rates of hospital-onset CRE and CephRE between period 1 and period 2 were also analyzed as the internal controls. 

### 2.7. Antimicrobial Consumption before and after the Enhancement of Infection Control Measures

The yearly overall antimicrobial consumption and the selected classes of antimicrobial agents, including carbapenems (ertapenem, meropenem, and imipenem/cilastatin), extended spectrum betalactam-betalactamase inhibitor (BLBI) combination (piperacillin/tazobactam, ticarcillin/clavulanate), cephalosporins (cefepime, cefoperazone, cefotaxime, ceftaroline, ceftazidime, and ceftriaxone), and intravenous and oral fluoroquinolones (ciprofloxacin, levofloxacin, and moxifloxacin) were analyzed and expressed as the DDD per 1000 patient days. The change in antimicrobial consumption between period 1 and period 2 was also analyzed. 

### 2.8. Statistical Analysis

Differences in the incidence rates of antimicrobial-resistant organisms and antimicrobial consumption were evaluated between period 1 (1 January 2007 to 31 December 2013) and period 2 (1 January 2014 to 31 December 2019) using interrupted time series, in particular segmented Poisson regression. At the time when the enhanced infection control measures were fully implemented (the year 2014), the change in the slope was investigated by including the corresponding term in the regression. Insignificant change was removed from the model. To facilitate the interpretation of results, in the case of significant change in the slope, the slopes of the two segments were presented (instead of the change). All statistical analyses were performed using IBM SPSS Statistics (version 26). A two-sided *p*-value of < 0.05 was considered statistically significant.

## 3. Results 

### 3.1. Trends of Antimicrobial-Resistant Organisms

The incidence rates of antimicrobial-resistant organisms, including CRAB, MRAB, CRE, and CephRE, expressed as the number of cases per 1000 patient days from 2007 to 2019, are shown in Figure 1. 

For all CRAB infections, the total number of infections per 1000 patient days increased by 22.7% per year [relative risk (RR): 1.227, 95% confidence interval (CI): 1.175–1.281, *p* < 0.001] in period 1, but decreased by 10.4% per year (RR: 0.896, 95% CI: 0.856–0.938, *p* < 0.001) in period 2. The number of clinical specimens cultured positive for *Acinetobacter baumannii*, which met the definitions of CRAB and MRAB, were comparable. The total number of MRAB infections per 1000 patient days increased by 14.6% per year (RR: 1.146, 95% CI: 1.111–1.182, *p* < 0.001) in period 1, but decreased by 9.7% per year (RR: 0.903, 95% CI: 0.869–0.938, *p* < 0.001) in period 2. 

For all CRE infections, the total number of infections per 1000 patient days increased by 10.8% per year (RR: 1.108, 95% CI: 1.081–1.135, *p* < 0.001) in both period 1 and period 2. There was no significant change in the trend of CRE infection between period 1 and period 2. For all CephRE infections, the total number of infections per 1000 patient days increased by 5.6% per year (RR: 1.056, 95% CI: 1.047–1.064, *p* < 0.001) in period 1, but decreased by 2.4% per year (RR: 0.976, 95% CI: 0.962–0.989, *p* < 0.001) in period 2. 

### 3.2. Compliance with Hand Hygiene and Directly Observed Hand Hygiene 

The hand hygiene compliance among healthcare workers increased from 23.2% (95% CI: 21.4–21.9%) (2007) to 75% (95% CI: 74.0–76.0%) in 2013, whereas the overall hand hygiene compliance among healthcare workers ranged from a minimum of 72.6% (95% CI: 71.9–73.4%) to a maximum of 79.5% (95% CI: 78.8–80.2%) between 2014 and 2019. The observed hand hygiene opportunities ranged from a minimum of 10,247 to a maximum of 14,340 between 2014 and 2019 (Figure 2). 

A systematic audit of directly observed hand hygiene was introduced in the third quarter of 2018 (during period 2). The compliance of the practice with the directly observed hand hygiene applied to the conscious hospitalized patients is illustrated in Table 1. The overall rate of compliance was consistently around 80%.

### 3.3. Hospital-onset Events before and after the Enhancement of Infection Control Measures

The incidence rates of hospital-onset infections due to antimicrobial-resistant organisms, including CRAB, MRAB, CRE, and CephRE, expressed as the number of cases per 1000 patient days from 2007 to 2019, are shown in Figure 3. The incidence rate of hospital-onset CRAB infection increased from 0.047 per 1000 patient days (2007) to the peak at 0.217 per 1000 patient days (2013), and decreased to 0.126 per 1000 patient days in 2019. 

For hospital-onset CRAB infections, the number of infections per 1000 patient days increased by 21.3% per year (RR: 1.213, 95% CI: 1.162–1.266, *p* < 0.001) in period 1. In period 2, a decrease of 9.8% per year was observed (RR: 0.902, 95% CI: 0.854–0.953, *p* < 0.001). Similarly, the number of hospital-onset MRAB infections increased by 13.0% per year (RR: 1.130, 95% CI: 1.095–1.166, *p* < 0.001) in period 1. In period 2, a decrease of 9.1% per year was observed (RR: 0.909, 95% CI: 0.868–0.953, *p* < 0.001). 

For hospital-onset CRE infections, the number of infections per 1000 patient days increased by 7.7% per year (RR: 1.077, 95% CI: 1.048–1.108, *p* < 0.001) in both period 1 and period 2. There was no significant change in the trend between the two periods. For hospital-onset CephRE infections, the number of infections increased by 3.3% per year (RR: 1.033, 95% CI: 1.024–1.042, *p* < 0.001) in period 1. In period 2, a decrease of 2.1% per year was observed (RR: 0.979, 95% CI: 0.963–0.995, *p* = 0.009).

### 3.4. Antimicrobial Consumption before and after the Enhancement of Infection Control Measures

The overall antimicrobial consumption and the selected classes of antimicrobial agents in the Queen Mary Hospital, expressed as DDD per 1000 patient days from 2007 to 2019, is shown in Figure 4. 

The consumption of all antimicrobial agents (DDD per 1000 patient days) increased by 2.1% per year (RR: 1.021, 95% CI: 1.018–1.024, *p* < 0.001) in both period 1 and period 2. There was no significant change in the trend between the two periods. For the selected antimicrobial classes, the carbapenems consumption (DDD per 1000 patient days) increased by 8.4% per year (RR: 1.840, 95% CI: 1.073–1.094, *p* < 0.001) in both period 1 and period 2. There was no significant change in the trend between the two periods. The extended spectrum BLBI combination consumption (DDD per 1000 patient days) increased by 11.2% per year (RR: 1.112, 95% CI: 1.102–1.122, *p* < 0.001) in period 1. In period 2, an increase of 3.8% per year was observed (RR: 1.038, 95% CI: 1.033–1.044, *p* < 0.001). The cephalosporins consumption (DDD per 1000 patient days) increased by 4.2% per year (RR: 1.042, 95% CI: 1.028–1.056, *p* < 0.001) in period 1. In period 2, an increase of 7.6% per year was observed (RR: 1.076, 95% CI: 1.056–1.097, *p* < 0.001). The fluoroquinolones consumption (DDD per 1000 patient days) increased by 4.6% per year (RR: 1.046, 95% CI: 1.031–1.062, *p* < 0.001) in period 1. In period 2, a decrease of 2.3% per year was observed (RR: 0.977, 95% CI: 0.967–0.987, *p* < 0.001).

## 4. Discussion

In this study, we have demonstrated the control of healthcare-associated CRAB infection, as evidenced by a significant reduction in the incidence rate of hospital-onset CRAB infection by means of enhanced infection control measures, in a regional hospital where the consumption of antimicrobial agents had paradoxically increased. In Hong Kong, CRAB and MRAB have been endemic in acute care settings [44,45] and emerging in long-term care facilities [46]. They have not only posed a burden of clinical infections in high-risk patient groups [47] but have also contributed to nosocomial transmission and outbreaks [48,49]. Antimicrobial stewardship is the primary intervention for preventing the emergence of antimicrobial-resistant organisms and protecting global health [50], but the sustainability of the stewardship program remains a great challenge in the healthcare settings, regardless of differences between resource-rich and resource-limited areas [51]. Our institution has been one of the pioneers in promoting the judicious use of antimicrobial agents in controlling antimicrobial-resistant organisms since the early 2000s [52,53]. However, the ongoing antimicrobial stewardship program was not able to revert the increasing trend of antimicrobial consumption, especially carbapenems, extended spectrum BLBI combination, and cephalosporins, as shown in our study. For instance, the carbapenems consumption (DDD per 1000 patient days) significantly increased by 8.4% per year during the study period. The extended spectrum BLBI combination and cephalosporins consumption rates also significantly increased by 11.2% and 4.2% per year, which correlated with a corresponding increase in the incidence rates of all CRAB infections and hospital-onset CRAB infections by 22.7% and 21.3% per year, respectively, in period 1. Similarly, the incidence rates of all MRAB infections and hospital-onset MRAB infections also significantly increased by 14.6% and 13.0%, respectively, in period 1. Therefore, we have to focus on the infection control and preventive measures to minimize the risk of the nosocomial transmission of antimicrobial-resistant organisms. 

There are numerous infection control measures for minimizing the risk of CRAB transmission in outbreak and non-outbreak settings, reported across Asia, Australia, Europe, and North America (Table 2). Staff education, patient isolation in single rooms or cohorting, environmental cleaning and disinfection, active surveillance culture upon admission and during hospitalization, daily chlorhexidine bathing, and even converting a multi-bed bay room to single rooms by renovation have been implemented. 

Compliance is the key for success in the implementation of any intervention for infection control. Hand hygiene is the cornerstone of infection control, but compliance with hand hygiene among healthcare workers was consistently less than 40% in different healthcare settings before the campaign based on using alcohol-based hand rub by WHO [54,55]. The choice of formulation of alcohol-based hand rub may also be of concern [56,57]. Therefore, we adopted the WHO formulation of alcohol-based hand rub for the hand hygiene campaign. At the baseline, compliance with hand hygiene among our healthcare workers was 23% in 2007. With the promotion of WHO guidelines on hand hygiene in healthcare workers, the overall hand hygiene compliance progressively increased to around 65% in 2009, and further increased to around 75% in 2011. However, the hand hygiene compliance has been static at around 75% since then, although a report of 100% hand hygiene compliance was achieved in some units during the COVID-19 pandemic [58]. Therefore, an innovative measure of directly observed hand hygiene was introduced to ensure compliance with the hand hygiene practice, which has not been implemented by the other centers (Table 2). 

Directly observed hand hygiene was initially implemented in the clinical areas with impending hospital outbreaks. When there were one or two patients with the nosocomial acquisition of epidemiologically important antimicrobial-resistant organisms, such as VRE [34], respiratory viruses such as metapneumovirus [31], and gastrointestinal viruses such as norovirus [59], alcohol-based hand rub was delivered to all healthcare workers by a designated member of staff in the clinical areas at 2- to 4-h intervals. The practice of directly observed hand hygiene should not replace the “Five Moments of Hand Hygiene” recommended by the WHO. It does not just serve to alert the healthcare workers of the impending hospital outbreaks, but also to disinfect the hands of staff and patients at regular intervals when the risk of outbreak in the clinical area is higher than usual. Directly observed hand hygiene has been shown to control and prevent the nosocomial transmission and outbreaks of different pathogens [31,32,33,34]. Therefore, the concept of directly observed hand hygiene was further promoted by healthcare workers to hospitalized patients. 

Patient participation or empowerment in hand hygiene is one of the key initiatives for facilitating multimodal hand hygiene promotion [60,61]. However, the implementation of patient empowerment in hand hygiene appears to be more complicated when the factors of the patients’ beliefs and perceptions are considered [62]. Our Chinese population is even more introvert and not willing to remind the healthcare workers to perform hand hygiene before touching them [37]. It is also challenging to build a culture of hand hygiene practice among patients by posting the education materials in the clinical areas [63]. To overcome these barriers, the concept of directly observed hand hygiene was modified to allow passive participation by the hospitalized patients. Before the meal and medication rounds, a designated member of staff is responsible for delivering 3 mL of alcohol-based hand rub to the conscious patients to reduce the risk of oral acquisition of antimicrobial-resistant organisms into the gastrointestinal tract. Directly observed hand hygiene should be the first priority in the infection control interventions, because it is easy to conduct and involves minimal cost. It only requires the deployment of a healthcare assistant to deliver alcohol-based hand rub to the conscious patients before breakfast, lunch, and dinner, as well as the three to four medication rounds per day. It may also increase the awareness of hand hygiene among the patients.

The implementation of directly observed hand hygiene before meal and medication rounds had successfully controlled the MRAB bacteremia in our hospital, as well as the territory-wide outbreak of VRE in our locality, after the first and second years of implementation, respectively [35,36]. The compliance with directly observed hand hygiene before meal and medication rounds was around 80%. Unlike hand hygiene audits among healthcare workers, which may be subjected to the Hawthorne effect, the audit of directly observed hand hygiene can be performed by interviewing three patients, with reasonable communication taking place in each ward at any time. Since directly observed hand hygiene is conducted at least five times per day, the risk of recall bias should be low. 

It is important to assess the effect of directly observed hand hygiene by investigating the change in incidence rates of antimicrobial-resistant organisms in period 2, especially since the consumption of carbapenems, extended spectrum BLBI combination, and cephalosporins were increasing. However, the fluoroquinolones consumption decreased significantly, by 2.3% per year, in period 2. This may have had an additive effect on the control of CRAB and MRAB infections, as the use of fluoroquinolones was found to be a risk factor for high bacterial load in patients with nasal and gastrointestinal colonization by MRAB [64]. The incidence rates of all CRAB infections and hospital-onset CRAB infections significantly decreased by 10.4% and 9.8% per year, respectively, after the implementation of the enhanced infection control measures in period 2. Similarly, the incidence rates of all MRAB infections and hospital-onset MRAB infections also significantly decreased by 9.7% and 9.1% per year, respectively, during the corresponding period. A similar trend of reduction in the incidence rates observed for both CRAB and MRAB infections was expected, because carbapenem resistance has become the surrogate marker for MRAB in recent years. The control of CRAB and MRAB infections were mainly attributed to the directly observed hand hygiene, the priority use of single-room isolation. In fact, the infection control nurses also regularly monitored the performance of environmental disinfection in the clinical areas and provided regular training to the cleaning staff during the study period [48]. 

Directly observed hand hygiene alone may not be enough to control all forms of antimicrobial-resistant organisms. With the analysis of the trend of CRE, which was used as an internal control, the incidence rates of all CRE infections and hospital-onset CRE infections significantly increased, by 10.8% and 7.7% per year, respectively, in both period 1 and period 2. Unlike infection control measures for CRAB and MRAB, where the priority use of single-room isolation was applied, cohort nursing was only applied to patients with CRE infection. The increasing trend of CRE infections may also correlate with the increasing prevalence of CRE carriage in the community [65]. The presence of CRE has been reported in livestock and vegetables [66,67,68]. At the same time, it is interesting to learn the outcome of another internal control, whereby the incidence rate of hospital-onset CephRE infections significantly decreased by 2.1% per year in period 2, even without the use of single rooms. This may reflect the net effects of directly observed hand hygiene on an antimicrobial-resistant organism which has been endemic in the community.

There were several limitations to this study. Firstly, this was a single-center descriptive study, without a control hospital to compare the changes in incidence rates of the overall hospital-onset antimicrobial-resistant organism infection before and after the enhancement of infection control measures, in light of an increasing trend of antimicrobial consumption, with the exception of fluoroquinolones. The interaction between the antimicrobial consumption and infection control measures may be more complicated, and it may require further exploration using mathematical models. Secondly, we only included the clinical specimens, and not the screening specimens collected for active surveillance purposes, in the calculation of incidence rates of antimicrobial-resistant organisms. The parameter of nosocomial transmission per 1000 colonization days was not analyzed as in the previous studies [69,70,71]. As this study was carried out over a long period from 2007 to 2019, the active surveillance protocol for antimicrobial-resistant organisms also changed over time [71,72,73]. Therefore, we adopted the laboratory-identified event reporting protocol defined by the NHSN of CDC [42]. Thirdly, the period of analysis did not include the COVID-19 pandemic because escalating infection control measures, including universal masking and directly observed environmental disinfection, were implemented, in addition to the practice of directly observed hand hygiene, which can affect the data interpretation [74,75,76,77].

## 5. Conclusions

Healthcare-associated CRAB can be controlled by directly observed hand hygiene-based infection control measures.

## Figures and Tables

**Figure 1 antibiotics-11-01076-f001:**
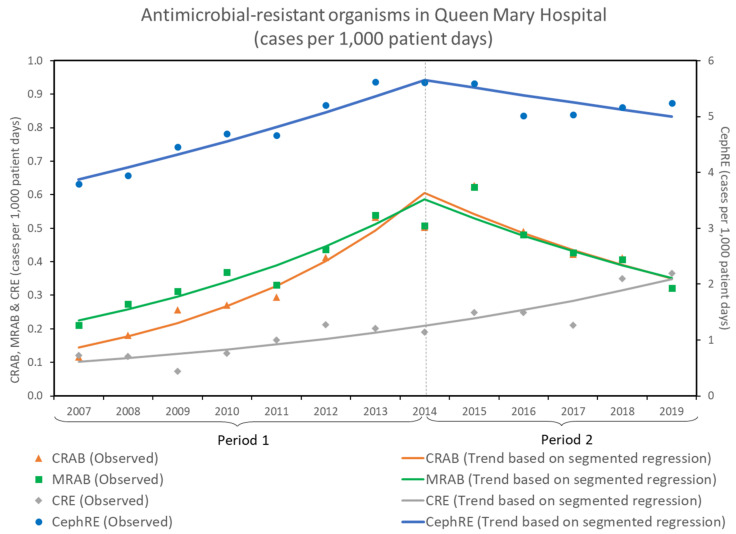
Antimicrobial-resistant organisms in Queen Mary Hospital from 2007 to 2019. CRAB, carbapenem-resistant *Acinetobacter baumannii*; CephRE, cephalosporin-resistant Enterobacterales; CRE, carbapenem-resistant Enterobacterales; MRAB, multidrug-resistant *Acinetobacter baumannii*. Remark: CRAB was defined as *Acinetobacter baumannii* which was non-susceptible (either resistant or intermediate) to either imipenem or meropenem being tested in our microbiology laboratory. MRAB was defined as *Acinetobacter baumannii* which was non-susceptible (either resistant or intermediate) to at least one agent in at least 3 antimicrobial classes of aminoglycosides, extended spectrum BLBI combination, carbapenems, cephalosporins, fluoroquinolones, and sulbactam [41]. CRE was defined as the microorganisms (*E. coli, Klebsiella species, and Enterobacter species*), under the order of Enterobacterales commonly cause infections in healthcare settings, non-susceptible (either resistant or intermediate) to either imipenem or meropenem [43]. CephRE was defined as the microorganisms (*E. coli, Klebsiella species, and Enterobacter species*) non-susceptible (either resistant or intermediate) to either cefepime, ceftazidime, or ceftriaxone in this study.

**Figure 2 antibiotics-11-01076-f002:**
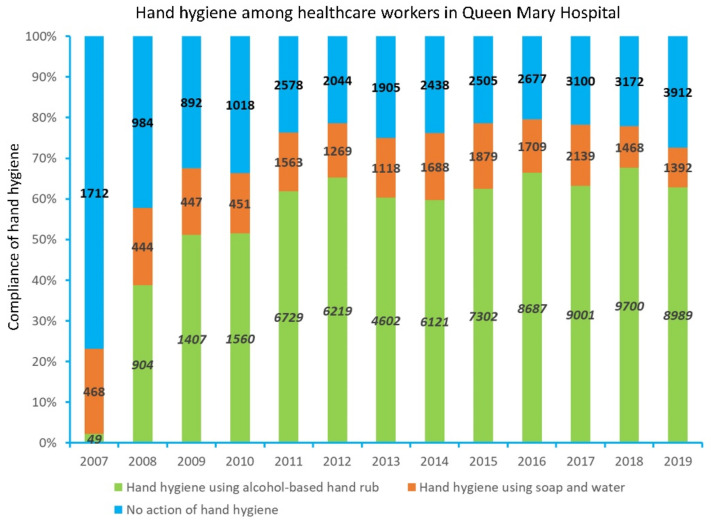
Hand hygiene compliance among healthcare workers in Queen Mary Hospital from 2007 to 2019. The numbers marked in the vertical bars represent the number of observed hand hygiene opportunities by infection control nurses. Non-mediated soap was used throughout the years of study.

**Figure 3 antibiotics-11-01076-f003:**
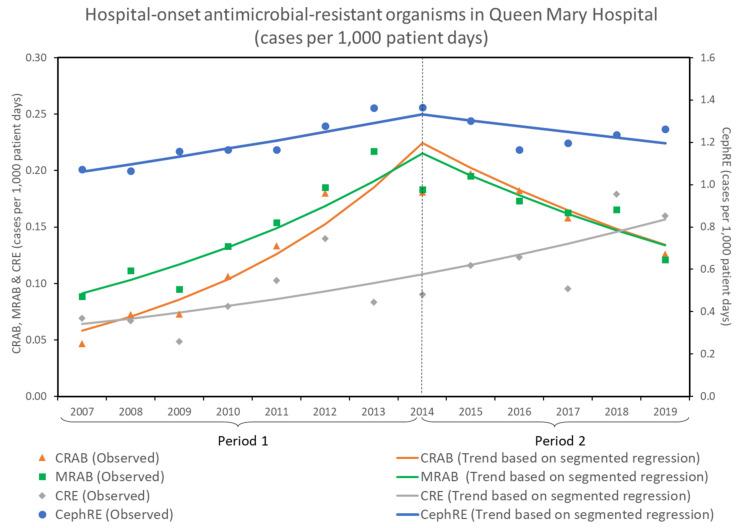
Hospital-onset antimicrobial-resistant organisms in Queen Mary Hospital from 2007 to 2019. CRAB, carbapenem-resistant *Acinetobacter baumannii*; CephRE, cephalosporin-resistant Enterobacterales; CRE, carbapenem-resistant Enterobacterales; MRAB, multidrug-resistant *Acinetobacter baumannii*.

**Figure 4 antibiotics-11-01076-f004:**
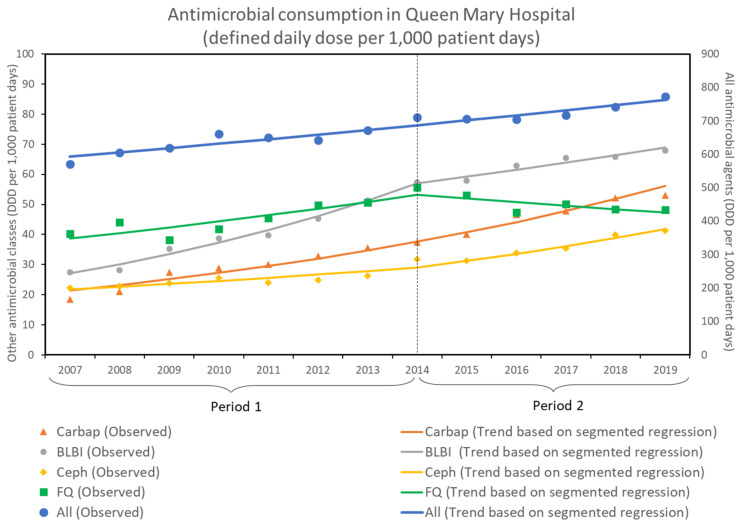
Antimicrobial consumption in Queen Mary Hospital before and after the enhancement of infection control measures. All, all antimicrobial agents; BLBI, extended spectrum betalactam-betalactamase inhibitor combination; Carbap, carbapenems; Ceph, cephalosporins; FQ; fluoroquinolones.

**Table 1 antibiotics-11-01076-t001:** Compliance with the practice of directly observed hand hygiene applied to the conscious hospitalized patients.

	Episodes of Audit	Episodes with Compliance	Percentage of Compliance	95% CI of the Percentage of Compliance
2018 3Q	636	508	79.9	(76.8–83.0)
2018 4Q	542	456	84.1	(81.1–87.2)
2019 1Q	474	391	82.5	(79.1–85.9)
2019 2Q	453	373	82.3	(78.8–85.9)
2019 3Q	479	391	81.6	(78.2–85.1)
2019 4Q	542	423	78.0	(74.6–81.5)

CI, confidence interval.

**Table 2 antibiotics-11-01076-t002:** Successful experiences of controlling carbapenem-resistant *Acinetobacter baumannii* (CRAB) in outbreak and non-outbreak periods.

No.	Year (Country) of Publication	Nature of Study (Setting)	Infection Control Measures	Ref
1	2010 (Korea)	Observational study during outbreak (ICU)	Enforcing contact precautions, environmental cleaning, and use of a closed-suctioning system	[15]
2	2010 (Australia)	Observational study during outbreak (ICU)	Single room isolation with contact precautions; using commercial oxidizing disinfectant with ICU closure for 3 days	[16]
3	2011 (USA)	Observational study during outbreak (NICU)	Active surveillance cultures of all infants, cohorting of affected infants and their nursing staff, contact isolation, environmental cleaning, and use of educational modules	[3]
4	2015 (Greece)	Observational study during outbreak (NICU)	Active surveillance (weekly stool samples), staff education, daily infection control audits and discontinuation of new admissions for 12 days	[4]
5	2018 (Israel)	Observational study during outbreak (ICU)	Unit closure for 3 days, environmental cleaning, hand hygiene interventions, and environmental culture	[17]
6	2021 (Italy)	Observational study during outbreak (ICU)	Enforcing hand hygiene, contact precautions to all patients, enhanced environmental sampling, and one-time application of a cycling radical environmental cleaning and disinfection	[18]
7	2014 (USA)	Quasi-experimental study (public hospital)	Weekly and systematic dissemination of the findings of infection control interventions	[19]
8	2014 (Korea)	Intervention study (tertiary hospital)	Onsite education and hand hygiene campaign in addition to cohorting, active surveillance, and environmental cleaning	[20]
9	2015 (Korea)	Intervention study (MICU) ^a^	Daily chlorhexidine bathing	[21]
10	2017 (Korea)	Intervention study (MICU)	Universal glove and gown use with daily chlorhexidine bathing for all patients in addition to surveillance cultures, contact precautions, and environmental cleaning	[22]
11	2019 (Japan)	Quasi-experimental study (ICU)	Active surveillance upon admission, weekly thereafter, and upon discharge	[23]
12	2020 (Thailand)	Intervention study (NICU) ^a^	Use of heat and moisture exchangers and sodium hypochlorite cleaning (5000 ppm in the NICU and 500 ppm in the environment)	[24]
13	2020 (Israel)	Intervention study (secondary-care hospital)	Maintaining a case registry of all CRAB patients, cohorting patients under strict contact isolation, using dedicated nursing staff and equipment, rigorous cleaning, education and close monitoring of hospital staff, and involvement of hospital management	[25]
14	2021 (Israel)	Intervention study (NSICU) ^a^	Wall painting using a water based acrylic paint following patient discharge and terminal cleaning with Sodium dichloroisocyanurate (sodium troclosene)	[26]
15	2021 (German)	Retrospective study (university hospital)	Single-room isolation and mandatory personal protective equipment (gloves, gowns, and surgical mask) for staff when caring for CRAB patients, and using disposable medical items	[27]
16	2022 (Korea)	Intervention study (MICU)	Renovated from a multi-bed bay room to single rooms for isolation of CRAB patients	[28]

ICU, intensive care unit; MICU, medical intensive care unit; NICU, neonatal intensive care unit; NSICU, neurosurgical intensive care unit: Ref, references. ^a^ using interrupted time series analysis.

## Data Availability

Data are available upon reasonable request.

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
