# Peer review of "Control of Healthcare-Associated Carbapenem-Resistant Acinetobacter baumannii by Enhancement of Infection Control Measures"

_antibiotics, 2022, doi:10.3390/antibiotics11081076_

Round 1
Reviewer 1 Report
Comments
Title:
The title of the manuscript is confusing, is necessary modified and most short.
Introduction
Lines 45 – 40: Global dissemination of eight ……..as illustrated by the molecular epidemiological analysis……..?. which?. The sentence is wrong and modified is required.
Lines 48 -49. “The nosocomial transmission and hospital outbreaks have been reported…..”: Where has been reported?, include this information.
Lines 51 - 53: I don’t understand the next sentence: “It is not unexpected to observe a higher clinical burden of CRAB in intensive care units because the likelihood of receiving broad-spectrum antimicrobial agents is higher in the critically ill patients”. What do the authors refer to with the following phrase "a higher clinical burden of CRAB", please explain or modify the sentence.
Lines 54 – 55: The next sentence “Antimicrobial consumption has been found to be correlated with the detection of CRAB in the hospitals [10-12], and also supported by the mathematical modelling study [13]”, can be improved.
Lines 61 – 65: In the sentence: “A five-component bundle of hand hygiene improvement, extended sampling at screening including the environment, universal contact precautions and a novel cycling radical environmental cleaning and disinfection procedure proved to be effective for eliminating CRAB spreading within the ICU” describe five components associated to hand hygiene; however, don’t described what the five components were. It is very risky to affirm that with these measures the dispersion of CRAB is eliminated; there is more bibliography that could contribute to this premise with a different panorama.
Material and Methods
2.1 Setting
Lines 83 – 88. The next sentence “This is a retrospective study on the epidemiology of healthcare associated CRAB, one of the antimicrobial resistant organisms, in Queen Mary Hospital, a university-affiliated teaching hospital of 1300-bed, providing a tertiary referral service for medical, surgical, orthopedics, pediatric, obstetrics and gynecology, oncology, solid organ and bone marrow transplantation in Hong Kong” could provide, i.e., eliminated “one of the antimicrobial resistant organisms”. Modified this sentence.
Lines 93 – 96. A reorganization of idea in the next sentence “Infection Control Team led by an infection control officer and a senior infection control nurse section is responsible for the planning and supervision of the infection control measures in the hospital, including the control of nosocomial transmission of antimicrobial resistant organisms” is required; also, is confusing.
The section of “Infection Control Measures for Antimicrobial Resistant Organisms” is very confusing, and is confusing and heavy to read. Authors are encouraged to better summarize and organize ideas.
Line 150: What do the authors mean by the term " immediate"
Lines 148 – 150. According to next sentence: “The gram-negative antimicrobial resistant organisms were selected for analysis in this study”, how many classes of antibiotics were considered to be resistant.
Lines 151 – 154. The next sentence: “CRE is defined as the microorganisms (E. coli, Klebsiella species, and Enterobacter species), under the order of Enterobacterales commonly cause infections in healthcare settings, non-susceptible (either resistant or immediate) to either imipenem or meropenem [30]” no is material and methods; therefore, could be eliminated.
This section: 2.4 Definition of Antimicrobial Resistant Organisms” must be removed
Of the manuscript and briefly included in introduction
Results
Line 213. Eliminated the word “Note:”
The different sections described in Results require organization again. In its present form it is not viable and is difficult to follow the aim of the work. As a suggestion for the authors, is review each of the sections in detail and write the ideas clearly and concisely.
Finally, major changes are required in the manuscript in its current form.
Author Response
Reply to Reviewer 1
Comments
Title:
The title of the manuscript is confusing, is necessary modified and most short.
Ans: Thank you for the comment. Together with the comment from Reviewer 4 that our infection control measure is mainly focusing on the directly observed hand hygiene, the manuscript title is changed from “Control of Healthcare Associated Carbapenem-resistant Acinetobacter baumannii Infection despite of an Increasing Trend of Antimicrobial Consumption” to “Control of Healthcare Associated Carbapenem-resistant Acinetobacter baumannii by Enhancement of Infection Control Measures”. The word court is reduced from 16 to 13.
Introduction
Lines 45 – 40: Global dissemination of eight ……..as illustrated by the molecular epidemiological analysis……..?. which?. The sentence is wrong and modified is required.
Ans: Thank you for the comment.
The original sentence is “Global dissemination of eight carbapenem-resistant lineages as illustrated by the molecular epidemiological analysis has demonstrated the success of epidemic spread in North and Latin America, Europe, Asia, South Africa and Australia [3].”
To avoid the confusion, we have deleted this sentence and its associated reference 3 in the revised manuscript.
Lines 48 -49. “The nosocomial transmission and hospital outbreaks have been reported…..”: Where has been reported?, include this information.
Ans: Thank you for the comment.
The original sentence is “Nosocomial transmission and hospital outbreaks have been reported, especially in the neonatal and adult intensive care units [4-7].”
And the references 4 to 7 in the original submission are as follows:
- McGrath, E.J.; Chopra, T.; Abdel-Haq, N.; Preney, K.; Koo, W.; Asmar, B.I.; Kaye, K.S. An outbreak of car-bapenem-resistant Acinetobacter baumannii infection in a neonatal intensive care unit: investigation and control. In-fect. Control. Hosp. Epidemiol. 2011, 32, 34-41.
- Tsiatsiou, O.; Iosifidis, Ε.; Katragkou, A.; Dimou, V.; Sarafidis, K.; Karampatakis, T.; Antachopoulos, C.; Orfanou, A.; Tsakris, A.; Drossou-Agakidou, V.; et al. Successful management of an outbreak due to carbapenem-resistant Acineto-bacter baumannii in a neonatal intensive care unit. Eur. J. Pediatr. 2015, 174, 65-74.
- Nhu, N.T.-K.; Lan, N.P.-H.; Campbell, J.I.; Parry, C.M.; Thompson, C.; Tuyen, H.T.; Hoang, N.V.-M.; Tam, P.T.-T.; Le, V.M.; Nga, T.V.-T.; et al. Emergence of carbapenem-resistant Acinetobacter baumannii as the major cause of ventila-tor-associated pneumonia in intensive care unit patients at an infectious disease hospital in southern Vietnam. J. Med. Microbiol. 2014, 63, 1386-1394.
- Ng, D.H.-L.; Marimuthu, K.; Lee, J.J.; Khong, W.X.; Ng, O.T.; Zhang, W.; Poh, B.F.; Rao, P.; Raj, M.D.-R.; Ang, B.; et al. Environmental colonization and onward clonal transmission of carbapenem-resistant Acinetobacter baumannii (CRAB) in a medical intensive care unit: the case for environmental hygiene. Antimicrob. Resist. Infect. Control. 2018, 7, 51.
For reference 4, it was reported from the USA. For reference 5, it was reported from Greece. For reference 6, it was reported from Vietnam. For reference 7, it was reported from Singapore.
We follow reviewer’s suggestion to revised this sentence in the revised manuscript.
Line 48-49:
Nosocomial transmission and hospital outbreaks, especially in the neonatal and adult intensive care units, have been reported in the United States, Greece, Vietnam, and Singapore.”
Lines 51 - 53: I don’t understand the next sentence: “It is not unexpected to observe a higher clinical burden of CRAB in intensive care units because the likelihood of receiving broad-spectrum antimicrobial agents is higher in the critically ill patients”. What do the authors refer to with the following phrase "a higher clinical burden of CRAB", please explain or modify the sentence.
Ans: Thank you for the comment. We have removed the phrase of “a higher clinical burden of CRAB”. The sentence is revised as follows:
Line 52-53:
“A high prevalence of CRAB in ICU could be attributed to the use of broad-spectrum anti-microbial agents in the critically ill patients”
Lines 54 – 55: The next sentence “Antimicrobial consumption has been found to be correlated with the detection of CRAB in the hospitals [10-12], and also supported by the mathematical modelling study [13]”, can be improved.
Ans: Thank you for the comment.
The sentence is revised as follows:
Line 54-57:
“The correlation between antimicrobial consumption and prevalence of CRAB has been observed in the hospitals. Mathematical models also suggest the existence of thresholds of antimicrobial consumption beyond which resistance would be triggered.”
Lines 61 – 65: In the sentence: “A five-component bundle of hand hygiene improvement, extended sampling at screening including the environment, universal contact precautions and a novel cycling radical environmental cleaning and disinfection procedure proved to be effective for eliminating CRAB spreading within the ICU” describe five components associated to hand hygiene; however, don’t described what the five components were. It is very risky to affirm that with these measures the dispersion of CRAB is eliminated; there is more bibliography that could contribute to this premise with a different panorama.
Ans: Thank you for the comment. We agree with reviewer’s suggestion to include more bibliography that could contribute to this premise with a different panorama. Therefore, we search the PubMed once again using the keyword of “control of carbapenem-resistant Acinetobacter baummannii” as title, there are 195 published articles as of 20 July 2022. Therefore, we conduct literature review of these 195 papers, and select 16 papers which are mainly focused on the endemic and outbreak settings due to carbapenem-resistant Acinetobacter baummannii. The infection control measures include the followings:
Choi WS, Kim SH, Jeon EG, Son MH, Yoon YK, Kim JY, Kim MJ, Sohn JW, Kim MJ, Park DW. Nosocomial outbreak of carbapenem-resistant Acinetobacter baumannii in intensive care units and successful outbreak control program. J Korean Med Sci. 2010 Jul;25(7):999-1004. doi: 10.3346/jkms.2010.25.7.999. Epub 2010 Jun 17. PMID: 20592889; PMCID: PMC2890899.
Doidge M, Allworth AM, Woods M, Marshall P, Terry M, O'Brien K, Goh HM, George N, Nimmo GR, Schembri MA, Lipman J, Paterson DL. Control of an outbreak of carbapenem-resistant Acinetobacter baumannii in Australia after introduction of environmental cleaning with a commercial oxidizing disinfectant. Infect Control Hosp Epidemiol. 2010 Apr;31(4):418-20. doi: 10.1086/651312. PMID: 20175684.
McGrath EJ, Chopra T, Abdel-Haq N, Preney K, Koo W, Asmar BI, Kaye KS. An outbreak of carbapenem-resistant Acinetobacter baumannii infection in a neonatal intensive care unit: investigation and control. Infect Control Hosp Epidemiol. 2011 Jan;32(1):34-41. doi: 10.1086/657669. Epub 2010 Nov 22. PMID: 21091204.
Tsiatsiou O, Iosifidis Ε, Katragkou A, Dimou V, Sarafidis K, Karampatakis T, Antachopoulos C, Orfanou A, Tsakris A, Drossou-Agakidou V, Roilides E. Successful management of an outbreak due to carbapenem-resistant Acinetobacter baumannii in a neonatal intensive care unit. Eur J Pediatr. 2015 Jan;174(1):65-74. doi: 10.1007/s00431-014-2365-8. Epub 2014 Jul 2. PMID: 24985124.
Ben-Chetrit E, Wiener-Well Y, Lesho E, Kopuit P, Broyer C, Bier L, Assous MV, Benenson S, Cohen MJ, McGann PT, Snesrud E, Levin PD. An intervention to control an ICU outbreak of carbapenem-resistant Acinetobacter baumannii: long-term impact for the ICU and hospital. Crit Care. 2018 Nov 21;22(1):319. doi: 10.1186/s13054-018-2247-y. PMID: 30463589; PMCID: PMC6249923.
Meschiari M, Lòpez-Lozano JM, Di Pilato V, Gimenez-Esparza C, Vecchi E, Bacca E, Orlando G, Franceschini E, Sarti M, Pecorari M, Grottola A, Venturelli C, Busani S, Serio L, Girardis M, Rossolini GM, Gyssens IC, Monnet DL, Mussini C. A five-component infection control bundle to permanently eliminate a carbapenem-resistant Acinetobacter baumannii spreading in an intensive care unit. Antimicrob Resist Infect Control. 2021 Aug 19;10(1):123. doi: 10.1186/s13756-021-00990-z. PMID: 34412693; PMCID: PMC8376111. [ref 17]
Munoz-Price LS, Carling P, Cleary T, Fajardo-Aquino Y, DePascale D, Jimenez A, Hughes M, Namias N, Pizano L, Kett DH, Arheart K. Control of a two-decade endemic situation with carbapenem-resistant Acinetobacter baumannii: electronic dissemination of a bundle of interventions. Am J Infect Control. 2014 May;42(5):466-71. doi: 10.1016/j.ajic.2013.12.024. PMID: 24773784.
Cho OH, Bak MH, Baek EH, Park KH, Kim S, Bae IG. Successful control of carbapenem-resistant Acinetobacter baumannii in a Korean university hospital: a 6-year perspective. Am J Infect Control. 2014 Sep;42(9):976-9. doi: 10.1016/j.ajic.2014.05.027. PMID: 25179329.
Chung YK, Kim JS, Lee SS, Lee JA, Kim HS, Shin KS, Park EY, Kang BS, Lee HJ, Kang HJ. Effect of daily chlorhexidine bathing on acquisition of carbapenem-resistant Acinetobacter baumannii (CRAB) in the medical intensive care unit with CRAB endemicity. Am J Infect Control. 2015 Nov;43(11):1171-7. doi: 10.1016/j.ajic.2015.07.001. Epub 2015 Aug 18. PMID: 26297525.
Hong J, Jang OJ, Bak MH, Baek EH, Park KH, Hong SI, Cho OH, Bae IG. Management of carbapenem-resistant Acinetobacter baumannii epidemic in an intensive care unit using multifaceted intervention strategy. Korean J Intern Med. 2018 Sep;33(5):1000-1007. doi: 10.3904/kjim.2016.323. Epub 2017 Nov 27. PMID: 29172401; PMCID: PMC6129627.
Yamamoto N, Hamaguchi S, Akeda Y, Santanirand P, Chaihongsa N, Sirichot S, Chiaranaicharoen S, Hagiya H, Yamamoto K, Kerdsin A, Okada K, Yoshida H, Hamada S, Oishi K, Malathum K, Tomono K. Rapid screening and early precautions for carbapenem-resistant Acinetobacter baumannii carriers decreased nosocomial transmission in hospital settings: a quasi-experimental study. Antimicrob Resist Infect Control. 2019 Jun 27;8:110. doi: 10.1186/s13756-019-0564-9. PMID: 31297191; PMCID: PMC6598269.
Thatrimontrichai A, Pannaraj PS, Janjindamai W, Dissaneevate S, Maneenil G, Apisarnthanarak A. Intervention to reduce carbapenem-resistant Acinetobacter baumannii in a neonatal intensive care unit. Infect Control Hosp Epidemiol. 2020 Jun;41(6):710-715. doi: 10.1017/ice.2020.35. PMID: 32131902.
Alon D, Mudrik H, Chowers M, Shitrit P. Control of a hospital-wide outbreak of carbapenem-resistant Acinetobacter baumannii (CRAB) using the Israeli national carbapenem-resistant Enterobacteriaceae (CRE) guidelines as a model. Infect Control Hosp Epidemiol. 2020 Aug;41(8):926-930. doi: 10.1017/ice.2020.158. Epub 2020 Jun 16. PMID: 32539881.
Dickstein Y, Eluk O, Warman S, Aboalheja W, Alon T, Firan I, Putler RKB, Hussein K. Wall painting following terminal cleaning with a chlorine solution as part of an intervention to control an outbreak of carbapenem-resistant Acinetobacter baumannii in a neurosurgical intensive care unit in Israel. J Infect Chemother. 2021 Oct;27(10):1423-1428. doi: 10.1016/j.jiac.2021.05.017. Epub 2021 May 31. PMID: 34083145.
Chhatwal P, Ebadi E, Schwab F, Ziesing S, Vonberg RP, Simon N, Gerbel S, Schlüter D, Bange FC, Baier C. Epidemiology and infection control of carbapenem resistant Acinetobacter baumannii and Klebsiella pneumoniae at a German university hospital: a retrospective study of 5 years (2015-2019). BMC Infect Dis. 2021 Nov 27;21(1):1196. doi: 10.1186/s12879-021-06900-3. PMID: 34837973; PMCID: PMC8627082.
Jung J, Choe PG, Choi S, Kim E, Lee HY, Kang CK, Lee J, Park WB, Lee S, Kim NJ, Choi EH, Oh M. Reduction in the acquisition rate of carbapenem-resistant Acinetobacter baumannii (CRAB) after room privatization in an intensive care unit. J Hosp Infect. 2022 Mar;121:14-21. doi: 10.1016/j.jhin.2021.12.012. Epub 2021 Dec 17. PMID: 34929231.
Therefore, we have deleted the original sentence
“A five-component bundle of hand hygiene improvement, extended sampling at screening including the environment, universal contact precautions and a novel cycling radical en-vironmental cleaning and disinfection procedure proved to be effective for eliminating CRAB spreading within the ICU [17].”
And revised the sentence as follows:
Line 60-66:
“Infection control measures other than antimicrobial stewardship program have been demonstrated to control the emerging CRAB in the hospitals during outbreak [3,4,14-17] and non-outbreak periods [18-27]. These measures include enforcing hand hygiene practice, education of hospital staff, single room isolation or cohorting of infected patients, environmental cleaning and disinfection, active surveillance culture upon admission and during hospitalization, daily chlorhexidine bathing, and renovation from a multi-bed bay room to single rooms.”
Material and Methods
2.1 Setting
Lines 83 – 88. The next sentence “This is a retrospective study on the epidemiology of healthcare associated CRAB, one of the antimicrobial resistant organisms, in Queen Mary Hospital, a university-affiliated teaching hospital of 1300-bed, providing a tertiary referral service for medical, surgical, orthopedics, pediatric, obstetrics and gynecology, oncology, solid organ and bone marrow transplantation in Hong Kong” could provide, i.e., eliminated “one of the antimicrobial resistant organisms”. Modified this sentence.
Ans: Thank you for the comment. We have eliminated “one of the antimicrobial resistant organisms” according to the suggestion.
Lines 93 – 96. A reorganization of idea in the next sentence “Infection Control Team led by an infection control officer and a senior infection control nurse section is responsible for the planning and supervision of the infection control measures in the hospital, including the control of nosocomial transmission of antimicrobial resistant organisms” is required; also, is confusing.
Ans: Thank you for the comment. We have made a major re-organization of this section suggested. This sentence is deleted in the revised version to avoid confusion.
The section of “Infection Control Measures for Antimicrobial Resistant Organisms” is very confusing, and is confusing and heavy to read. Authors are encouraged to better summarize and organize ideas.
Ans: Thank you for the comment. The section of “Infection Control Measures for Antimicrobial Resistant Organisms” has been re-organized. It is divided into two different sections.
2.2 Promotion of Hand Hygiene and Directly Observed Hand hygiene
2.3 Infection Control Measures for Antimicrobial Resistant Organisms
We believe the content is better summarize and organize in the revised version.
Line 150: What do the authors mean by the term " immediate"
Ans: Thank you for the question. The term “immediate: should be read as “intermediate”. It means the microorganism is intermediate resistant to an antimicrobial agent.
This term is under the section of “2.4 Definition of Antimicrobial Resistant Organisms”, and this section is deleted according to reviewer’s suggestion.
Lines 148 – 150. According to next sentence: “The gram-negative antimicrobial resistant organisms were selected for analysis in this study”, how many classes of antibiotics were considered to be resistant.
Ans: Thank you for the question. We selected carbapenem-resistant Acinetobacter baumannii (CRE), multidrug-resistant Acinetobacter baumannii (MRAB), carbapenem-resistant Enterobacterales (CRE), and cephalosporin-resistant Enterobacterales (CephRE) for analysis.
CRAB is defined as Acinetobacter baumannii which is non-susceptible to either imipenem or meropenem. CRE is defined as the Enterobacterales (E. coli, Klebsiella species, and Enterobacter species) which is non-susceptible to either imipenem or meropenem. CephRE is defined as the Enterobacterales (E. coli, Klebsiella species, and Enterobacter species) which is non-susceptible to either cefepime, ceftazidime, or ceftriaxone. MRAB is defined as Acinetobacter baumannii which is non-susceptible to at least one agent in at least 3 antimicrobial classes of aminoglycosides, betalactam-betalactamase inhibitor combination, carbapenems, cephalosporins, fluoroquinolones, and sulbactam.
These definitions were included in the section “2.4 Definition of Antimicrobial Resistant Organisms” in the original submission.
We have followed reviewer’s suggestion to remove the section “2.4 Definition of Antimicrobial Resistant Organisms” in the revised version.
Lines 151 – 154. The next sentence: “CRE is defined as the microorganisms (E. coli, Klebsiella species, and Enterobacter species), under the order of Enterobacterales commonly cause infections in healthcare settings, non-susceptible (either resistant or immediate) to either imipenem or meropenem [30]” no is material and methods; therefore, could be eliminated.
Ans: Thank you for the comment. We have removed this sentence together with the whole section of “2.4 Definition of Antimicrobial Resistant Organisms” in the revised version according to reviewer’s suggestion.
This section: 2.4 Definition of Antimicrobial Resistant Organisms” must be removed
Of the manuscript and briefly included in introduction
Ans: Thank you for the comment. This section 2.4 Definition of Antimicrobial Resistant Organisms” is removed from the revised version.
Results
Line 213. Eliminated the word “Note:”
Ans: Thank you for the question. The word “Note” is eliminated in the revised manuscript.
The different sections described in Results require organization again. In its present form it is not viable and is difficult to follow the aim of the work. As a suggestion for the authors, is review each of the sections in detail and write the ideas clearly and concisely.
Ans: Thank you for the comment. We greatly appreciate the suggestion from reviewer, especially reminding us to change the manuscript title, so that we can focus the aim of our work. In this study, we would like to demonstrate the implementation of directly observed hand hygiene, together with other infection control measures, to control the healthcare associated CRAB, even though the antimicrobial consumption was in an increasing trend.
In the section “3.1 Trend of Antimicrobial Resistant Organisms”
We have revised Figure 1 for better illustration of the trend of antimicrobial resistant organisms before and after implementation of enhancement of infection control measures using segmented regression analysis
For a better presentation, we are going to re-organize the results by removing the content
“Figure 2: Percentage change of antimicrobial resistant organisms in Queen Mary Hospital with reference to the baseline data on 2007.”, which may not be directly relevant to our aim of study.
In the section “3.2 Compliance of Hand Hygiene and Directly Observed Hand Hygiene”
We have re-organized the content for better illustration of compliance hand hygiene first (revised Figure 2), and then to present the compliance of directly observed hand hygiene in Table 1.
In the section “3.3 Hospital-onset Events before and after Enhancement of Infection Control Measures”
We have revised the name of section as “3.3 Hospital-onset Events before and after Enhancement of Infection Control Measures”
We have also revised Figure 3 for better illustration of the Hospital-onset antimicrobial resistant organisms before and after enhancement of infection control measures using segmented regression analysis
In the section “3.4 Antimicrobial consumption before and after Enhancement of Infection Control Measures”
We have revised the name of section as “3.4 Antimicrobial consumption before and after Enhancement of Infection Control Measures”
We have revised Figure 4 for better illustration of the antimicrobial consumption before and after enhancement of infection control measures using segmented regression analysis.
For a better presentation, we are going to re-organize the results by removing the content
“Figure 6: Percentage change of antimicrobial consumption per year with reference to the base-line data on 2007.”, which may not be directly relevant to our aim of study.
Finally, major changes are required in the manuscript in its current form.
Ans: Thank you for the comment. We have made major changes to improve the manuscript according to reviewer’s suggestion.
Reviewer 2 Report
The authors conducted a retrospective analysis on the incidence rate of hospital-onset carbapenem-resistant Acinetobacter baumannii (CRAB) infection from period 1 (2007 to 2013) to period 2 (2014 to 2019), where enhanced infection control measures including directly observed hand hygiene before meal and medication rounds to conscious patients, and priority use of single room isolation were implemented in period 2. The study has been well conducted, following some comments and minor revisions.
Introduction
Line 49: insert acronym ICU for Intensive Care Units and sostitute it with ICU in line 51 and 52.
Lines 73-76: If it is a work already published insert bibliography if it refers to the results of the work, in this section the purpose should be highlighted, not the result.
Materials and Methods
Line 89: Enter the complete start and end dates for period 1 and period 2.
Line 156: Enter the full meaning of the acronym MRAB.
Results:
Figure 2: divide the results according to the first (2007-2013) and the second observation period (2014-2019), in accordance with what is written in the text (lines 220-224), both for CRAb, CRE and CephRE.
Indicate in the figure the data of MRAB infections
Lines 236- 245: About the data regarding hand cleaning, I would indicate what type of alcohol based hand rub was used, especially in the discussion: recent data indicate a different antibacterial effect compared to the composition of alcohol based hand rub (10.3390 / gels8020087). Comment in the discussion indicating the bibliography: 10.3390 / gels8020087, 10.3390 / ijerph18126252.
Figure 4: highlight in the figure period 1 and period 2 as indicated in the text.
Indicate MRAb infections data in the figure
Line 275-277: for CRE infections if from 2007 to 2019 there is an increase of 7.7% per year, please explain why there is no significant change in the year 2014.
Figure 5 and Figure 6: highlight in the figure period 1 and period 2 as stated in the text .
Author Response
Reply to Reviewer 2
The authors conducted a retrospective analysis on the incidence rate of hospital-onset carbapenem-resistant Acinetobacter baumannii (CRAB) infection from period 1 (2007 to 2013) to period 2 (2014 to 2019), where enhanced infection control measures including directly observed hand hygiene before meal and medication rounds to conscious patients, and priority use of single room isolation were implemented in period 2. The study has been well conducted, following some comments and minor revisions.
Introduction
Line 49: insert acronym ICU for Intensive Care Units and substitute it with ICU in line 51 and 52.
Ans: Thank you for the comment. We have revised the manuscript accordingly.
Lines 73-76: If it is a work already published insert bibliography if it refers to the results of the work, in this section the purpose should be highlighted, not the result.
Ans: Thank you for the comment.
The original sentence is
“With the implementation of directly observed hand hygiene-based infection control measures for 5 years, we demonstrated the effect of this intervention on the incidence rate of hospital-onset CRAB even though there was a significant increasing trend of antimicrobial consumption.”
The sentence is revised as follows:
Line 74-77:
“With the implementation of directly observed hand hygiene-based infection control measures for 5 years, we would like to demonstrate the effect of this intervention on the incidence rate of hospital-onset CRAB even though there was a significant increasing trend of antimicrobial consumption.”
Materials and Methods
Line 89: Enter the complete start and end dates for period 1 and period 2.
Ans: Thank you for the comment.
The original sentence is
The study period is divided into two parts, where period 1 (2007 to 2013) is the baseline and period 2 (2014 to 2019) is the intervention by enhancement in infection control measures.
The sentence is revised as follows:
Line 85-88:
The study period is divided into two parts, where period 1 (1 January 2007 to 31 December 2013) is the baseline and period 2 (1 January 2014 to 31 December 2019) is the intervention by enhancement in infection control measures.
Line 156: Enter the full meaning of the acronym MRAB.
Ans: Thank you for the comment.
The full meaning of the acronym MRAB is multidrug-resistant Acinetobacter baumannii.
In Line 156 (original submission), it is the second time to describe the term “MRAB.
The full meaning of the acronym MRAB is stated in Line 119-120 in the first submission.
Results:
Figure 2: divide the results according to the first (2007-2013) and the second observation period (2014-2019), in accordance with what is written in the text (lines 220-224), both for CRAb, CRE and CephRE.
Indicate in the figure the data of MRAB infections
Ans: Thank you for the comment.
We have revised Figure 1 for better illustration of the trend of antimicrobial resistant organisms before and after enhancement of infection control measures using segmented regression analysis.
For a better presentation, we are going to re-organize the results by removing the content
“Figure 2: Percentage change of antimicrobial resistant organisms in Queen Mary Hospital with reference to the baseline data on 2007.”, in the revised version.
Lines 236- 245: About the data regarding hand cleaning, I would indicate what type of alcohol based hand rub was used, especially in the discussion: recent data indicate a different antibacterial effect compared to the composition of alcohol based hand rub (10.3390 / gels8020087). Comment in the discussion indicating the bibliography: 10.3390 / gels8020087, 10.3390 / ijerph18126252.
Ans: Thank you for the comment. We have been using World Health Organization formulation of alcohol-based hand rub since 2007.
This information is added in the section “2.2 Promotion of Hand Hygiene and Directly Observed Hand hygiene” under Materials and Methods in the revised version.
Line 93-96:
“WHO formulation I (ethanol 80% v/v, glycerol 1.45% v/v, hydrogen peroxide 0.125% v/v), and formulation II (isopropyl alcohol 75% v/v, glycerol 1.45% v/v, hydrogen peroxide 0.125% v/v) alcohol-based hand rub were promoted to our healthcare workers according to “My Five Moments for Hand Hygiene”.”
We have also cited these two references as reference number 56 and 57 in the discussion as suggested by reviewer.
d'Angelo I, Provenzano R, Florio E, Pagliuca C, Mantova G, Scaglione E, Vitiello M, Colicchio R, Salvatore P, Ungaro F, Quaglia F, Miro A. Alcohol-Based Hand Sanitizers: Does Gelling Agent Really Matter? Gels. 2022 Jan 29;8(2):87. doi: 10.3390/gels8020087. PMID: 35200468; PMCID: PMC8871833.
Booq RY, Alshehri AA, Almughem FA, Zaidan NM, Aburayan WS, Bakr AA, Kabli SH, Alshaya HA, Alsuabeyl MS, Alyamani EJ, Tawfik EA. Formulation and Evaluation of Alcohol-Free Hand Sanitizer Gels to Prevent the Spread of Infections during Pandemics. Int J Environ Res Public Health. 2021 Jun 9;18(12):6252. doi: 10.3390/ijerph18126252. PMID: 34207817; PMCID: PMC8296100.
Figure 4: highlight in the figure period 1 and period 2 as indicated in the text.
Indicate MRAb infections data in the figure
Ans: Thank you for the comment.
We have revised the Figure 4 (it is now renamed as Figure 3) according to the suggestion.
Period 1 and period 2 are indicated. Data of MRAB infection is also included.
Line 275-277: for CRE infections if from 2007 to 2019 there is an increase of 7.7% per year, please explain why there is no significant change in the year 2014.
Ans: Thank you for the comment.
No significant change in the trend means the 7.7% increase per year applied to both periods, covering 2007 to 2019.
To avoid any potential misunderstanding, we have revised the sentence as follows:
Line 274-276:
“For hospital-onset CRE infection, the number of infections per 1,000 patient days in-creased by 7.7% per year (RR: 1.077, 95% CI: 1.048 - 1.108, p <0.001) in both Period 1 and Period 2. There was no significant change in such trend between the two periods”
Figure 5 and Figure 6: highlight in the figure period 1 and period 2 as stated in the text.
Ans: Thank you for the comment.
Figure 5 is revised and renamed as Figure 4. The labels of period 1 and period 2 are added.
We follow the suggestion from another reviewer, Figure 6 is removed in the revised version.
Reviewer 3 Report
This is a retrospective study aimed to investigate the effect of enhanced infecytion control measures in the incidence of hospital CRAB in a tertiary referral centre in China. This is an interesting manuscript and all limitations are also addressed in a proper way. There are several minor comments that refer to syntax and editing.
Introduction: use the abbreviation ICU after first use
Line55: correct to: “, which was also supported by a mathematical study”
Line 56-59: please rephrase the meaning is confusing
Line 68: was carried out, who delivered……
Line 78: delete “in the trend”
Line 89: is the intervention period with enhancement
Line 94: was responsible
Line 106: ambassadors
Line 111: nurses.Three..
Line 115: two out of three
Line 125: CRAB
Line 125,126,127 : positive for
Line 135: was assigned
Line 136: episodes
Line 140: delete “were retrieved”
Line 141: was defined
Line 149: was defined…was susceptible
2.4: Use the past tense for the whole paragraph
Line 166: as a surrogate marker for multi-drug resistant A.baumanii
Line 175: was defined
Line 178: use recorded instead of counted
Line 179: were expressed
Line 192: the change in
Line 201: were presented
Line 208: are shown
Line 210: is shown
Line 236: compliance in
Line 239: delete was
Line 241: delete was
Line 245: is illustrated
Line 260: are shown
Line 285 : at …
Line 286,287: is shown
Discussion:
Line 311: CRAB,…
Line 314: agents, was paradoxically ….
Line 314: have instead of has
Line 321: Our institution was one of the pioneers…
Line 323: ongoing
Line 327: delete was
Line 334: for success….for infection control
Line 338: delete only
Author Response
Reply to Reviewer 3
This is a retrospective study aimed to investigate the effect of enhanced infection control measures in the incidence of hospital CRAB in a tertiary referral centre in China. This is an interesting manuscript and all limitations are also addressed in a proper way. There are several minor comments that refer to syntax and editing.
Introduction: use the abbreviation ICU after first use
Ans: Thanks, and we have made this amendment accordingly.
Line55: correct to: “, which was also supported by a mathematical study”
Ans: Thank you for the comment.
We have revised the sentence in the revised version.
Line 55-57:
“Mathematical models also suggest the existence of thresholds of antimicrobial consumption beyond which resistance would be triggered.”
Line 56-59: please rephrase the meaning is confusing
Ans: Thank you for the comment.
The original sentence is
“The introduction of antimicrobial stewardship program contributed to a reduction in the use of several broad-spectrum antibiotics was associated with a significant impact on reducing the incidence of CRAB in the hospital [13], as well as for the outbreak control [14].”
The sentence is revised as follows:
Line 57-59:
“Antimicrobial stewardship program was associated with a significant impact on reducing the incidence of CRAB in the hospital [13], as well as contributed to the outbreak control [14].”
Line 68: was carried out, who delivered……
Ans: Thanks, and we have made this amendment accordingly.
Line 78: delete “in the trend”
Ans: Thanks, and we have made this amendment accordingly.
Line 89: is the intervention period with enhancement
Ans: Thanks, and we have made this amendment accordingly.
Line 94: was responsible
Ans: Thanks, and this sentence was deleted in the revised version.
Line 106: ambassadors
Ans: Thanks, and we have made this amendment accordingly.
Line 111: nurses.Three..
Ans: Thanks, and we have made this amendment accordingly.
Line 115: two out of three
Ans: Thanks, and we have made this amendment accordingly.
Line 125: CRAB
Ans: Thanks, and we have made this amendment accordingly.
Line 125,126,127 : positive for
Ans: Thanks, and we have made this amendment accordingly.
Line 135: was assigned
Ans: Thanks, and we have made this amendment accordingly.
Line 136: episodes
Ans: Thanks, and we have made this amendment accordingly.
Line 140: delete “were retrieved”
Ans: Thanks, and we have made this amendment accordingly.
Line 141: was defined
Ans: Thanks, and we have made this amendment accordingly.
Line 149: was defined…was susceptible
Ans: Thanks, and this sentence was deleted in the revised version.
2.4: Use the past tense for the whole paragraph
Ans: Thanks, and whole paragraph was deleted in the revised version.
Line 166: as a surrogate marker for multi-drug resistant A.baumanii
Ans: Thanks, and we have made this amendment accordingly.
Line 175: was defined
Ans: Thanks, and we have made this amendment accordingly.
Line 178: use recorded instead of counted
Ans: Thanks, and we have made this amendment accordingly.
Line 179: were expressed
Ans: Thanks, and we have made this amendment accordingly.
Line 192: the change in
Ans: Thanks, and we have made this amendment accordingly.
Line 201: were presented
Ans: Thanks, and we have made this amendment accordingly.
Line 208: are shown
Ans: Thanks, and we have made this amendment accordingly.
Line 210: is shown
Ans: Thanks, and this sentence was deleted in the revised version.
Line 236: compliance in
Ans: Thanks, and we prefer to keep the sentence “compliance of hand hygiene”
Line 239: delete was
Ans: Thanks, and we have made this amendment accordingly.
Line 241: delete was
Ans: Thanks, and we have made this amendment accordingly.
Line 245: is illustrated
Ans: Thanks, and we have made this amendment accordingly.
Line 260: are shown
Ans: Thanks, and we have made this amendment accordingly.
Line 285 : at …
Ans: Thanks, and we have made this amendment accordingly.
Line 286,287: is shown
Ans: Thanks, and we have made this amendment accordingly.
Discussion:
Line 311: CRAB,…
Ans: Thanks, and we have made this amendment accordingly.
Line 314: agents, was paradoxically ….
Ans: Thanks, and we have made this amendment accordingly.
Line 314: have instead of has
Ans: Thanks, and we have made this amendment accordingly.
Line 321: Our institution was one of the pioneers…
Ans: Thanks, and we have made this amendment accordingly.
Line 323: ongoing
Ans: Thanks, and we have made this amendment accordingly.
Line 327: delete was
Ans: Thanks, and we have made this amendment accordingly.
Line 334: for success….for infection control
Ans: Thanks, and we have made this amendment accordingly.
Line 338: delete only
Ans: Thanks, and we have made this amendment accordingly.
Reviewer 4 Report
To authors:
The presented data is not new. I have some comments below:
1) The provided data is not a new finding. Can authors either come up with additional novelty aspects? If not applicable, please provided reviewed of previous works in Asia and try to contrast or compare other works with their study to make it a significant contribution.
2) Can authors clarify wording “intensified IC” in their manuscript? In the current version, the authors focused their efforts on hand hygiene mainly.
3) Previous works suggested that environmental cleaning is a significant part of IC for MDR-A. baumannii. Are there any intensified effort to enhance environmental cleaning?
4) Are there any attempt to perform source control using CHG bath or selective gut decontamination? These interventions listed in 3 and 4 may justified wording intensified.
5) In statistical analysis, authors mentioned about segmented regression analysis from time series. This is an important component. However, in Figure, I did not see analysis using segmented regression. Can authors analysis their outcomes using segmented regression analysis.
6) Discussion should be focused on key components in comparison with other components that have been successfully implemented particularly in Asia. This may make this manuscript become more unique.
7) How do you prioritize the interventions?
8) Limitations should include single center, the lack of control hospital, maturation to mean, hawthorn effect for HH, some interventions were implemented at different time frame.
9) To be able to really know that this is the effect from intervention, it is important to perform segmented regression analysis.
Author Response
Reply to Reviewer 4
To authors:
The presented data is not new. I have some comments below:
1) The provided data is not a new finding. Can authors either come up with additional novelty aspects? If not applicable, please provided reviewed of previous works in Asia and try to contrast or compare other works with their study to make it a significant contribution.
Ans: Thank you for the comment. We agree that our manuscript on the “control of carbapenem-resistant Acinetobacter baummannii” is not new. When we use this term “control of carbapenem-resistant Acinetobacter baummannii” as title in PubMed search, there are 195 published articles as of 20 July 2022. Therefore, we follow the reviewer’s suggestion to conduct literature review of these 195 papers, and select 16 papers which are mainly focused on the endemic and outbreak settings due to carbapenem-resistant Acinetobacter baummannii. A table of literature review of these 16 papers is prepared in our revised manuscript in order to facilitate the comparison of the intervention strategies among these 16 papers, which are not just limited from Asia, but also from Australia, Europe, and North America. It is noted that none of previously published work included the concept of directly observed hand hygiene in their interventions. It appears that our infection control measure with focus on directly observed hand hygiene before meal and medication rounds for all conscious hospitalized patients has additional novelty in the literature.
The Successful experience in controlling carbapenem-resistant Acinetobacter baumannii in outbreak and non-outbreak settings is summarized in a newly prepared Table 2 (Line 339) in the revised version.
2) Can authors clarify wording “intensified IC” in their manuscript? In the current version, the authors focused their efforts on hand hygiene mainly.
Ans: Thank you for the comment. We did not use the term “intensified IC” in our manuscript. However, we used the term “enhanced infection control measures” emphasize the implementation of directly observed hand hygiene before meal and medication rounds to conscious hospitalized patients, as well as the priority use of single room isolation for patient with CRAB infection. During the baseline period, we did not practice directly observed hand hygiene and the utilization of single room was not given to patients infected with CRAB. Therefore, we would like to keep the term “enhanced infection control measures” in the revised manuscript.
3) Previous works suggested that environmental cleaning is a significant part of IC for MDR-A. baumannii. Are there any intensified effort to enhance environmental cleaning?
Ans: Thank you for the comment. We did not intensify the environmental cleaning. The hospital environment is cleaned and disinfected once daily. For single rooms and cohort areas with MDROs including MDRA, it would be cleaned and disinfected twice daily. Since the experience of environmental cleaning and surveillance for MDRA was reported previously, we have cited our paper (reference 48) in the discussion.
Cheng VCC, Wong SC, Chen JHK, So SYC, Wong SCY, Ho PL, Yuen KY. Control of multidrug-resistant Acinetobacter baumannii in Hong Kong: Role of environmental surveillance in communal areas after a hospital outbreak. Am J Infect Control. 2018 Jan;46(1):60-66. doi: 10.1016/j.ajic.2017.07.010. Epub 2017 Sep 8. PMID: 28893447.
To supplement the effort in the environmental cleaning, we have added a sentence in the revised discussion as follows:
Line 409-411:
“In fact, the infection control nurses were regularly monitored the performance of environ-mental disinfection in the clinical areas, and provided regularly training to the cleaning staff during the study period.”
4) Are there any attempt to perform source control using CHG bath or selective gut decontamination? These interventions listed in 3 and 4 may justified wording intensified.
Ans: Thank you for the comment. We did not attempt to perform source control using chlorhexidine bath or selective gut decontamination. We agree with reviewer’s comment that the word “intensified” should not be used. Therefore, we only use the term “enhanced infection control measures” in our manuscript.
5) In statistical analysis, authors mentioned about segmented regression analysis from time series. This is an important component. However, in Figure, I did not see analysis using segmented regression. Can authors analysis their outcomes using segmented regression analysis.
Ans: Thank you for the comment. All the models are from segmented regression analysis, and the relevant statistical analysis shown in the RESULTS are from segmented regression analysis. We have revised Figure 1, Figure 4, and Figure 5 to make the result from segmented regression analysis visible.
6) Discussion should be focused on key components in comparison with other components that have been successfully implemented particularly in Asia. This may make this manuscript become more unique.
Ans: Thank you for the comment. We have added the Infection Control Measures in the control of carbapenem-resistant Acinetobacter baumannii not just in Asia, but other parts of the world, in the DISCUSSION, and also prepared a Table 2 (Line 339) to summarize the interventions in the revised manuscript.
Therefore, an innovative measure of directly observed hand hygiene was introduced to ensure the compliance of hand hygiene practice, which has not been implemented by the other centers as illustrated in Table 2.
7) How do you prioritize the interventions?
Ans: Thank you for the comment. I would prioritize directly observed hand hygiene as the first intervention, because it is easy to conduct and involve minimal cost. It only requires to deploy a healthcare assistant to deliver alcohol-based hand rub to the conscious patients before breakfast, lunch, and dinner, as well as the 3 to 4 times of medication rounds per day. With the implementation of directly observed hand hygiene, it may increase the awareness of hand hygiene of the patients.
We have added this sentence in the revised discussion.
Line 381-386:
“Directly observed hand hygiene should be the first priority in the infection control measures, because it is easy to conduct and involves minimal cost. It only requires to deploy a healthcare assistant to deliver alcohol-based hand rub to the conscious patients before breakfast, lunch, and dinner, as well as the 3 to 4 times of medication rounds per day. It may also increase the awareness of hand hygiene of the patients.”
8) Limitations should include single center, the lack of control hospital, maturation to mean, hawthorn effect for HH, some interventions were implemented at different time frame.
Ans: Thank you for the comment. We have addressed the limitation in the last paragraph of discussion as follows:
Line 427-430:
“There are several limitations in this study. First, this is a single-center descriptive study without a control hospital to compare the change in incidence rates of the overall and hospital-onset antimicrobial resistant organisms before and after enhancement of infection control measures,”
Line 436-438:
“As this study included a long period from 2007 to 2019, the focus of infection control measures may change with time.”
For the Hawthorne effect of hand hygiene, it is not applicable to the audit of directly observed hand hygiene, because the audit of directly observed hand hygiene performed by interviewing three patients with reasonable communication in each ward, instead of observing the delivery of alcohol-based hand rub to conscious patients by the healthcare assistance.
We have emphasized this point in the discussion as follows:
Line 391-395:
“Unlike hand hygiene audit among healthcare workers, which may be subjected to Hawthorne effect, audit of directly observed hand hygiene could be performed by interviewing three patients with reasonable communication in each ward any time. Since directly observed hand hygiene is conducted at least 5 times per day, the risk of recall bias should be low.”
9) To be able to really know that this is the effect from intervention, it is important to perform segmented regression analysis.
Ans: Thank you for the comment. Please be reassured that segmented regression analysis is applied to study the effect from intervention in this manuscript.
Reviewer 5 Report
I have evaluated the manuscript (Antibiotics-1816615) titled “Control of Healthcare Associated Carbapenem-resistant Acinetobacter baumannii Infection despite of an Increasing Trend of Antimicrobial Consumption” by Cheng and coworkers, and a retrospective analysis on the incidence rate of hospital-onset carbapenem-resistant Acinetobacter baumannii (CRAB) infection in Queen Mary Hospital has been done. All standard methods were used for this research. I found this article interesting for the readers and followed the journal Antibiotics’ scope. The author needs to discuss figures 1 and 2 in detail. The author needs to comment on the effect of the composition of alcohol-based hand soap/soap water for this study.
I would recommend the article could be published in Antibiotics after minor corrections.
Author Response
Reply to Reviewer 5
Comments and Suggestions for Authors
I have evaluated the manuscript (Antibiotics-1816615) titled “Control of Healthcare Associated Carbapenem-resistant Acinetobacter baumannii Infection despite of an Increasing Trend of Antimicrobial Consumption” by Cheng and coworkers, and a retrospective analysis on the incidence rate of hospital-onset carbapenem-resistant Acinetobacter baumannii (CRAB) infection in Queen Mary Hospital has been done. All standard methods were used for this research. I found this article interesting for the readers and followed the journal Antibiotics’ scope. The author needs to discuss figures 1 and 2 in detail. The author needs to comment on the effect of the composition of alcohol-based hand soap/soap water for this study.
Ans: Thank you for your comment.
We have the major revision according to the comment from another 4 reviewers.
For the original Figure 1 and Figure 2, we have made major revision by revising the presenting format to make the segmented regression analysis visible. The change of incidence rate of CRAB, MRAB, CRE, and CephRE in relation to the enhanced infection control measures are discussed.
The original version of Figure 2 was deleted according to the comment of other reviewer.
The composition of alcohol-based hand rub is added in the revised version.
Line 93-96:
WHO formulation I (ethanol 80% v/v, glycerol 1.45% v/v, hydrogen peroxide 0.125% v/v), and formulation II (isopropyl alcohol 75% v/v, glycerol 1.45% v/v, hydrogen peroxide 0.125% v/v) alcohol-based hand rub were promoted to our healthcare workers according to “My Five Moments for Hand Hygiene”.
We also mentioned the use of non-mediated soap in the legend of revised Figure 2.
Line 1076:
“Non-mediated soap was used throughout the years.”
I would recommend the article could be published in Antibiotics after minor corrections.
Round 2
Reviewer 1 Report
None
Reviewer 4 Report
No more comments